# Distinct Subtypes of Hepatorenal Syndrome and Associated Outcomes as Identified by Machine Learning Consensus Clustering

**DOI:** 10.3390/diseases11010018

**Published:** 2023-01-27

**Authors:** Supawit Tangpanithandee, Charat Thongprayoon, Pajaree Krisanapan, Michael A. Mao, Wisit Kaewput, Pattharawin Pattharanitima, Boonphiphop Boonpheng, Wisit Cheungpasitporn

**Affiliations:** 1Division of Nephrology and Hypertension, Department of Medicine, Mayo Clinic, Rochester, MN 55905, USA; 2Chakri Naruebodindra Medical Institute, Faculty of Medicine Ramathibodi Hospital, Mahidol University, Samut Prakan 10540, Thailand; 3Division of Nephrology, Department of Internal Medicine, Faculty of Medicine Thammasat University, Pathum Thani 12120, Thailand; 4Division of Nephrology, Department of Internal Medicine, Thammasat University Hospital, Pathum Thani 12120, Thailand; 5Division of Nephrology and Hypertension, Department of Medicine, Mayo Clinic, Jacksonville, FL 32224, USA; 6Department of Military and Community Medicine, Phramongkutklao College of Medicine, Bangkok 10400, Thailand; 7Department of Medicine, Division of Nephrology, University of Washington, Seattle, WA 98195, USA

**Keywords:** acute kidney injury, AKI, cirrhosis, clustering, hepatorenal syndrome, HRS, machine learning

## Abstract

Background: The utilization of multi-dimensional patient data to subtype hepatorenal syndrome (HRS) can individualize patient care. Machine learning (ML) consensus clustering may identify HRS subgroups with unique clinical profiles. In this study, we aim to identify clinically meaningful clusters of hospitalized patients for HRS using an unsupervised ML clustering approach. Methods: Consensus clustering analysis was performed based on patient characteristics in 5564 patients primarily admitted for HRS in the National Inpatient Sample from 2003–2014 to identify clinically distinct HRS subgroups. We applied standardized mean difference to evaluate key subgroup features, and compared in-hospital mortality between assigned clusters. Results: The algorithm revealed four best distinct HRS subgroups based on patient characteristics. Cluster 1 patients (n = 1617) were older, and more likely to have non-alcoholic fatty liver disease, cardiovascular comorbidities, hypertension, and diabetes. Cluster 2 patients (n = 1577) were younger and more likely to have hepatitis C, and less likely to have acute liver failure. Cluster 3 patients (n = 642) were younger, and more likely to have non-elective admission, acetaminophen overdose, acute liver failure, to develop in-hospital medical complications and organ system failure, and to require supporting therapies, including renal replacement therapy, and mechanical ventilation. Cluster 4 patients (n = 1728) were younger, and more likely to have alcoholic cirrhosis and to smoke. Thirty-three percent of patients died in hospital. In-hospital mortality was higher in cluster 1 (OR 1.53; 95% CI 1.31–1.79) and cluster 3 (OR 7.03; 95% CI 5.73–8.62), compared to cluster 2, while cluster 4 had comparable in-hospital mortality (OR 1.13; 95% CI 0.97–1.32). Conclusions: Consensus clustering analysis provides the pattern of clinical characteristics and clinically distinct HRS phenotypes with different outcomes.

## 1. Introduction

Hepatorenal syndrome (HRS) is a reversible functional kidney injury that occurs in patients with advanced cirrhosis or fulminant hepatic failure [1,2]. It is a life-threatening condition that has an incidence as high as 32 percent among those with advanced liver disease [3,4,5,6,7,8,9]. HRS is characterized by a significant decrease in glomerular filtration rate (GFR) in the absence of other causes of kidney injury, exclusively in patients with ascites [10,11,12]. Severe renal vasoconstriction is the hallmark of HRS, which is caused by underlying pathophysiology of portal hypertension and systemic inflammation [2].

HRS has poor prognosis outcomes, with a mortality rate of up to 80% within two weeks and almost 90% within a month [13,14]. The high model of end-stage liver disease (MELD) score, sepsis, extrahepatic organ failure, and the high severity of acute kidney injury (AKI) are widely acknowledged as predictors of high mortality in patients with HRS [1,13,15,16,17,18,19,20]. Recently, a large study on the US national inpatient sample (NIS) database reported that older age, alcohol use, coagulopathy, and neurological disorder, and the need for mechanical ventilation are also associated with higher hospital mortality, whereas liver transplantation, transjugular intrahepatic portosystemic shunt, and abdominal paracentesis were associated with lower hospital mortality [21].

Although previous research has focused on identifying risk factors for developing HRS in patients with advanced liver disease to facilitate early detection and management, data on the utilization of machine learning (ML) approaches for patients with HRS are limited. The applicability of artificial intelligence (AI) and ML in clinical care has attracted considerable attention and numerous investigations [22,23,24,25,26,27,28,29,30,31,32,33,34,35,36,37]. The principle of ML is the capacity of computers to recognize and evaluate trends and patterns in data without being explicitly programmed. Consensus clustering is an unsupervised machine-learning approach that identifies unique data patterns by looking for similarities and heterogeneities among huge categories of data variables and grouping them into clinically relevant clusters [38,39,40,41,42]. Recent research has shown that discrete subtypes defined by ML consensus clustering method can predict varied clinical outcomes [42,43]. An improved understanding of the diverse phenotypes of HRS patients may help clinicians to discover personalized methods to enhance outcomes for susceptible people in this group [44].

In this study, we aim to identify clinically meaningful clusters of hospitalized patients for HRS using an unsupervised ML clustering approach. We then assessed the outcomes of these distinct clusters.

## 2. Materials and Methods

### 2.1. Study Population

The National Inpatient Sample (NIS) database is the largest all-payer database of hospital admissions in the United States. It comprises the unweighted data of more than 7 million hospital admissions annually from a 20% stratified sample of over 4000 hospitals in the United States and represents 95% of hospital admission nationwide. The information contains patient demographics, principal diagnoses, and procedural codes. The database does not include individual patients, but it contains a single inpatient admission. Institutional review board approval was obtained (IRB number—22-011097). Data in the NIS is publicly available and de-identified.

We used NIS data from 2003 to 2014 to construct a cohort of hospital admissions primarily for hepatorenal syndrome. We identified the diagnosis of hepatorenal syndrome based on the International Classification of Diseases, Ninth Edition (ICD-9) diagnosis code of 572.4.

### 2.2. Data Collection

Patient and admission characteristics data were abstracted to identify clinically distinct HRS clusters. Patient characteristics included age, sex, race, liver diseases, comorbidities, hospital events, organ system dysfunction, treatments, and Do-Not-Resuscitate (DNR) orders. Admission characteristics included elective vs. non-elective admission, and weekday vs. weekend admission. Outcome was in-hospital mortality.

### 2.3. Clustering Analysis

We applied an unsupervised ML approach to consensus clustering in order to identify clinical phenotypes of hospitalized patients for HRS. We performed consensus clustering analysis on the whole study population. Then, we initially assessed the distribution and missingness in phenotyping variables. Subsequently, non-normal data were z-score normalized. We subsequently applied clustering using the consensus cluster algorithm. The algorithm begins by subsampling a proportion of items and a proportion of features from a data matrix. Each subsample is then partitioned into groups (k) by a user-specified clustering algorithm. This process is repeated for a specified number of times. Pairwise consensus values, defined as ‘the proportion of clustering runs in which two items are grouped together’, are calculated and stored in a consensus matrix (CM) for each cluster. Clustering settings used were as follows: maximum number of clusters, 10; number of iterations, 100; subsampling fraction, 0.8; clustering algorithm, K-means; Euclidean distance [45]. We used a pre-specified subsampling parameter of 80% with 100 iterations and designated the number of potential clusters (k) range from 2 to 10 to avoid delivering an excessive number of clusters that would not be clinically meaningful. The optimal number of clusters was identified by evaluating the CM heat map, cumulative distribution function (CDF), cluster-consensus plots in the within-cluster consensus scores, and the proportion of ambiguously clustered pairs (PAC) [46,47]. The within-cluster consensus score (between 0 and 1) is defined as the average consensus value for all pairs of individuals belonging to the similar cluster [47]. A value closer to 1 indicates better cluster stability [47]. PAC, ranging between 0 and 1, is calculated as the proportion of all sample pairs with consensus values falling within the predetermined boundaries [46]. A value closer to 0 represents better cluster stability. This study’s detailed consensus clustering algorithms are provided in the Appendix A. Workflow of Consensus Clustering is shown in Figure 1.

### 2.4. Statistical Analysis

Following the identification of the clusters of HRS patients, we tested the differences between the assigned clusters. We evaluated patient characteristics and outcomes among the clusters utilizing the analysis of variance (ANOVA) test for continuous variables and the Chi-squared test for categorical variables. We identified the clusters’ key features by a standardized mean difference with a set cut-off of >0.3. We estimated the odds ratio (OR) for in-hospital mortality of each cluster using cluster 2 as the reference group, because cluster 2 was associated with the lowest mortality. We did not adjust for patient characteristics because these characteristics were utilized to identify clusters through unsupervised ML.

All cluster derivation analyses were performed using R, version 4.0.3 (RStudio, Inc., Boston, MA, USA), with the packages of ConsensusClusterPlus (version 1.46.0) (Bioconductor Open-Source Software for Bioinformatics) for consensus clustering analysis. CPU is a 4-core Intel Core i7-950 running at 3.07 GHz. All analyses were two-tailed, and *p*-value < 0.05 was considered statistically significant.

## 3. Results

Consensus clustering analysis was performed in 5564 hospital admissions, primarily for HRS in the NIS database from 2003 to 2014. The CDF plot displays the consensus distributions for each cluster (Figure 2A). The delta area plot demonstrates the relative change in the area under the CDF curve (Figure 2B). The largest changes in the area were found between k = 3 and k = 5, at which point the relative increase in the area became noticeably smaller. As demonstrated in the CM heatmap (Figure 2C, Appendix A), the ML algorithm revealed cluster 1 and cluster 3 with clear boundaries, indicating good cluster stability over repeated iterations. The mean cluster consensus score was comparable between a scenario of two or four clusters (Figure 3A). Favorable low PACs were demonstrated for two and four clusters (Figure 3B). Utilizing baseline variables at hospital admission, the consensus clustering analysis identified four clusters that best represented the data pattern of hospitalized HRS patients.

There were 1617 (29%) patients in cluster 1, 1577 (28%) patients in cluster 2, 642 (12%) in cluster 3, and 1728 (31%) patients in cluster 4. The patient characteristics were different among the four assigned clusters (Table 1). The plot of standardized mean difference in Figure 4 reveals the key features of each cluster. Cluster 1 patients were older (mean age 72 years) and more likely to have non-alcoholic fatty liver disease, hypertension, diabetes, congestive heart failure, atrial fibrillation, and coronary artery disease. Cluster 2 patients were younger (mean age 53 years), more likely to have hepatitis C, and less likely to have acute liver failure. Cluster 3 patients were younger (mean age 54 years) and more likely to have non-elective hospital admission, acetaminophen overdose, acute liver failure, gastrointestinal (GI) bleeding, cardiac arrest, metabolic acidosis, sepsis, circulatory, respiratory and hematological failure, and require renal replacement therapy (RRT), mechanical ventilation, blood transfusion, and nutritional support. Cluster 4 patients were younger (mean age 54 years) and more likely to have alcoholic cirrhosis and to smoke.

Thirty-three percent of patients died in hospitals. In-hospital mortality was 34% in cluster 1, 25% in cluster 2, 70% in cluster 3, and 27% in cluster 4 (*p* < 0.001). In-hospital mortality was higher in cluster 3 (OR 7.03; 95% CI 5.73–8.62) and cluster 1 (OR 1.53; 95% CI 1.31–1.79), as compared to cluster 2, while cluster 4 had comparable in-hospital mortality (OR 1.13; 95% CI 0.97–1.32).

## 4. Discussion

Four distinct clusters of hospitalized patients with HRS were identified in this study. Cluster 1, which accounts for 29% of all patients, is comprised of elderly patients with comorbidities including non-alcoholic fatty liver disease (NAFLD), hypertension, diabetes, congestive heart failure, atrial fibrillation, and coronary artery disease. Cluster 2, which accounts for 28%, is predominantly comprised of hepatitis C patients. Cluster 3, which accounts for 12%, is comprised primarily of patients with acute liver failure and acetaminophen overdose. This group of patients has significantly more in-hospital complications, including gastrointestinal bleeding, cardiac arrest, metabolic acidosis, sepsis, circulatory, respiratory, and hematological failure, requiring RRT, mechanical ventilation, blood transfusion, and nutritional support. Thirty-one percent of all patients in cluster 4 consisted of individuals with alcoholic cirrhosis and smoking.

Characteristics of patients in cluster 3 had the worst in-hospital mortality outcome among all clusters, given that they experienced the most significant number of multiple organ failures while hospitalized, including respiratory failure that required mechanical ventilation, circulatory failure leading to cardiac arrest, hematological failure that led to bleeding and required blood transfusion, and metabolic acidosis or renal impairment that required RRT. Recent evidence demonstrates that HRS patients with severe AKI requiring RRT carry a very high mortality rate, ranging from 60 to 80% within 28 days after the initiation of RRT. Also, nearly half of the patients in cluster 3 developed sepsis as a complication during admission, with sepsis being a well-known risk factor for in-hospital mortality ranging from 24 to 39% [48]. Moreover, it is noted that all of the patients in cluster 3 were admitted in an emergency condition, mostly due to acute liver failure and acetaminophen overdose. Since acute liver failure is associated with a mortality rate as high as 85% [49,50], this could explain why this cluster had such a high mortality rate. Even though acetaminophen-induced acute liver failure has a relatively low mortality rate of 27% in the general population [51], the incidence of death in patients with underlying liver disease is still unknown.

Cluster 1 also had significantly higher in-hospital mortality as compared to cluster 2. We hypothesize that age, specific comorbidities, and etiology of liver disease may play an essential role in the poor prognosis. Patients in cluster 1 had a mean age of 72.3 ± 8.4 years, which was significantly older than those in cluster 2, who had a median age of 53.0 ± 8.3 years, aligning with evidence that hospital mortality in patients with HRS is related to advanced age [21]. Moreover, certain comorbidities, particularly cardiovascular diseases, are widely recognized as risk factors for sudden cardiac death [52], and patients in cluster 1 had the highest number of cardiovascular risks, including hypertension, diabetes, hyperlipidemia, chronic kidney disease, coronary artery disease, atrial fibrillation, and congestive heart failure. Cluster 1 has the highest proportion of female patients. It has been reported that older patients with reduced GFR, especially females, have a higher prevalence of metabolic syndromes [53]. Therefore, the presence of these underlying conditions and advanced age may be a significant obstacle to liver transplantation, resulting in a greater mortality rate. For the etiology of liver disease, acute liver failure and NAFLD account for most of the causes of liver disease in this cluster. Even while acute liver failure is associated with a higher death rate, as we have discussed, the relationship between NAFLD and mortality remains uncertain. Future research is required to investigate this correlation in order to improve outcomes.

Given that cluster 4 primarily consisted of alcoholic cirrhosis, it is somewhat surprising that this group had a similar mortality rate to cluster 2 and had even better outcomes than clusters 1 and 3. It is possible that cluster 4 patients had the highest proportion of GI bleeding of all clusters, suggesting that GI bleeding may be a major cause of hospital admission, not from existing renal impairment. Since GI bleeding is an acute condition that needs hospitalization, early diagnosis and management of acute kidney injury from HRS are straightforward and may lead to better outcomes. For cluster 2, patients had the lowest mortality rate among all clusters. The explanations were that patients in this cluster had the highest proportion of elective admission and the lowest comorbidities. It is noted that half of these patients were hepatitis C-related cirrhosis. Studies show that an extrahepatic manifestation of liver diseases such as chronic hepatitis B virus is glomerulonephritis with a poor prognosis in adults [54,55]. Viruses can cause kidney abnormalities in a number of ways, each of which is frequently suggestive of a specific infection. The pathogenesis could be caused by circulating or in situ immune complexes, viral proteins or an inflammatory factor, direct effects of the virus on glomerular cells, tubulointerstitial injury, hemodynamic disruption in the case of septic shock caused by disseminated intravascular coagulation, rhabdomyolysis, or even antiviral medication [56]. Generally, the progression of hepatitis C develops gradually over time. Ten to twenty percent of individuals with hepatitis C will develop cirrhosis and other forms of chronic liver disease [56]. Glomerulonephritis, which is frequently accompanied by membranoproliferative changes, has been noted more frequently in HCV infection than in HBV infection [56]. However, data on the impact of hepatitis C on mortality in HRS patients are limited and thus require future studies.

This study had a few limitations. Due to the nature of this cohort study, there is a lack of detail specific to the exact causes of death, and there is a lack of code in the NIS database that distinguishes patients with HRS type 1 from those with type 2. Despite the fact that type 1 HRS is a rapidly progressive functional renal failure and type 2 HRS is a milder gradual decline in renal function, we were unable to determine the HRS type. Therefore, the ICD-9 code 572.4 was chosen to identify the diagnosis of all types of HRS. In addition, future studies are required to additionally validate this machine learning in different datasets of HRS patients. Lastly, while this unsupervised ML clustering approach can identify distinct phenotypes of HRS patients with differing posttransplant outcomes, ML clustering algorithms have limitations that do not directly generate risk prediction for each individual. Following identification of these four distinct clusters, we found that there are associated risk factors that may play important roles in in-hospital mortality among HRS patients, especially multiple organ failures, respiratory failure requiring mechanical ventilation, circulatory failure leading to cardiac arrest, hematological failure requiring blood transfusion, metabolic acidosis, and renal impairment requiring RRT as demonstrated in Cluster 3. Other risk factors on in-hospital mortality are age and cardiovascular comorbidities, which can be found in Cluster 1 patients. Future studies assessing the utilization of supervised ML prediction models for outcomes among patients with HRS are needed. Nevertheless, the findings from this ML clustering approach provide additional understanding towards individualized medicine for patients with HRS.

To the best of our knowledge, this is the first unsupervised ML approach that specifically targets patients with HRS. Consensus clustering analysis without human intervention synthesized the pattern of four clinically distinct HRS subgroups with different outcomes, highlighting opportunities to further early identify high-risk HRS patients. There is a need for additional research into the early management of HRS patients in order to improve survival.

## 5. Conclusions

In this cohort study, the analysis of the NIS database using an unsupervised machine learning clustering approach identified four clinically distinct clusters of patients with HRS, which were found to have different mortality outcomes. An improved understanding of these phenotypes may identify strategies for improving survival in HRS patients.

## Figures and Tables

**Figure 1 diseases-11-00018-f001:**
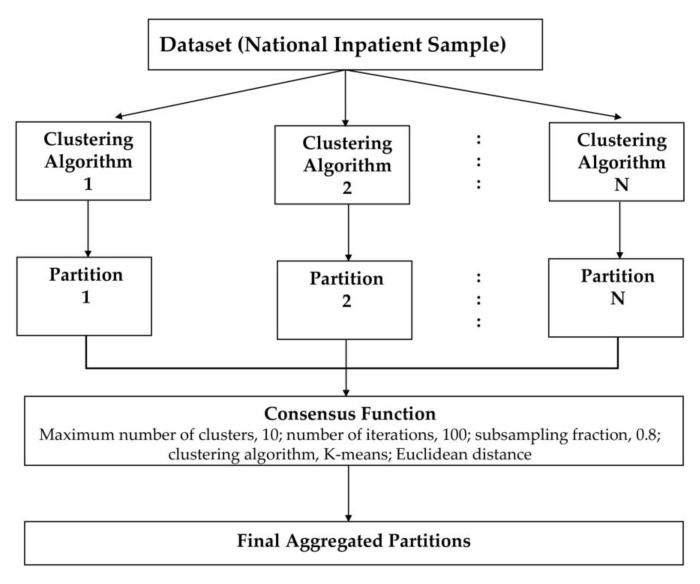
Workflow of Consensus Clustering.

**Figure 2 diseases-11-00018-f002:**
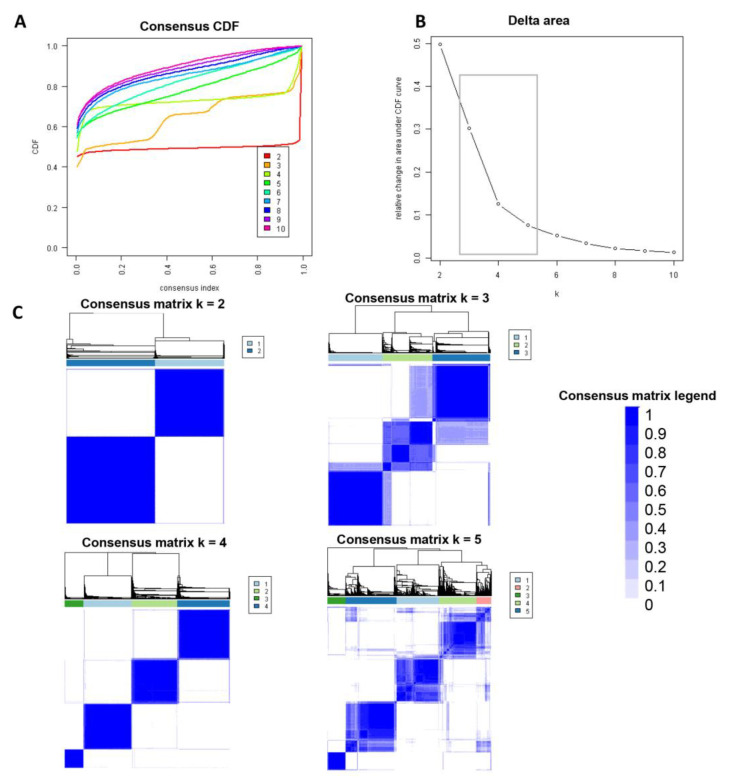
(**A**) CDF plot demonstrating consensus distributions for each cluster (k); (**B**) Delta area plot representing the relative changes in the area under the CDF curve. (**C**) Consensus matrix heat map displaying consensus values on a white to blue color scale of each cluster.

**Figure 3 diseases-11-00018-f003:**
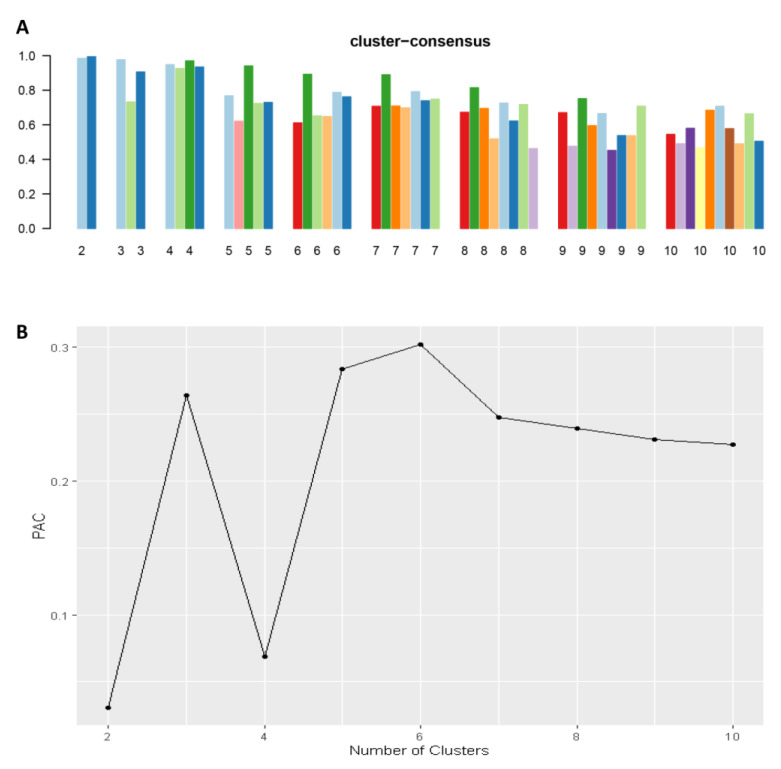
(**A**) The mean cluster consensus score was comparable between a scenario of two or four clusters. (**B**) Favorable low PACs were demonstrated for two and four clusters.

**Figure 4 diseases-11-00018-f004:**
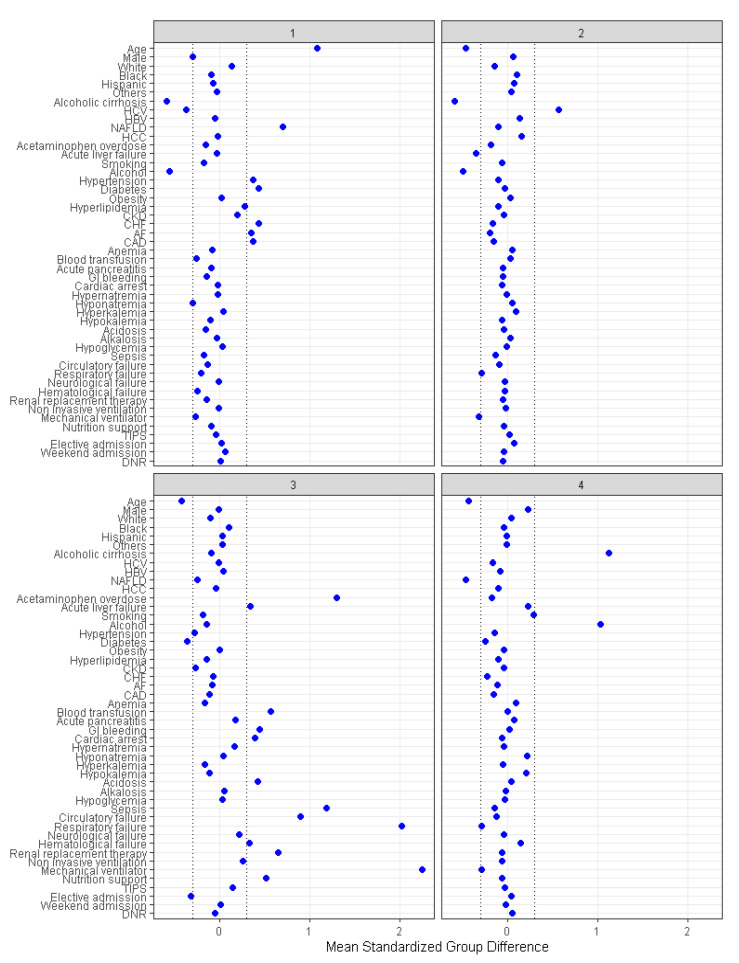
The standardized differences across four clusters for each baseline parameter. AF: Atrial fibrillation, CAD: Coronary artery disease, CKD: Chronic kidney disease, CHF: Chronic heart failure, DNR: Do not resuscitate, HBV: Hepatitis B virus, HCC: Hepatocellular carcinoma, HCV: Hepatitis C virus, NAFLD: Nonalcoholic fatty liver disease, TIPS: Transjugular intrahepatic portosystemic shunts.

**Table 1 diseases-11-00018-t001:** Clinical characteristics according to clusters of hepatorenal syndrome patients.

	All	Cluster 1	Cluster 2	Cluster 3	Cluster 4	*p*-Value
N	5564	1617	1577	642	1728	
Male sex	3533 (64)	791 (49)	1050 (67)	402 (63)	1290 (75)	<0.001
Age (years)	58.9 ± 12.4	72.3 ± 8.4	53.0 ± 8.3	53.6 ± 10.2	53.5 ± 9.5	<0.001
Race - White - Black - Hispanic - Other	4221 (76) 472 (8) 583 (10) 288 (5)	1320 (82) 93 (6) 134 (8) 70 (4)	1102 (70) 177 (11) 201 (13) 97 (6)	459 (72) 73 (11) 73 (11) 37 (6)	1340 (78) 129 (7) 175 (10) 84 (5)	<0.001
Weekend admission	1109 (20)	361 (22)	287 (18)	131 (20)	330 (19)	0.02
Elective admission	518 (9)	157 (10)	182 (12)	0 (0)	179 (10)	<0.001
Liver diseases - Alcoholic cirrhosis - Hepatitis B - Hepatitis C - NALFD - Hepatocellular carcinoma - Acetaminophen overdose - Acute liver failure	2081 (37) 149 (3) 1208 (22) 1067 (19) 343 (6) 205 (4) 2246 (40)	144 (9) 29 (2) 104 (6) 756 (47) 89 (6) 12 (1) 628 (39)	137 (9) 77 (5) 709 (45) 240 (15) 157 (10) * 363 (23)	211 (33) 21 (3) 137 (21) 59 (9) 33 (5) 181 (28) 364 (57)	1589 (92) 22 (1) 258 (15) 12 (1) 64 (4) * 891 (52)	<0.001 <0.001 <0.001 <0.001 <0.001 <0.001 <0.001
Comorbidities - Smoking - Alcohol use - Hypertension - Diabetes mellitus - Obesity - Hyperlipidemia - Chronic kidney disease - Congestive heart failure - Atrial fibrillation - Coronary artery disease	446 (8) 1796 (32) 1738 (31) 1184 (21) 340 (6) 352 (6) 1617 (29) 679 (12) 400 (7) 467 (8)	52 (3) 98 (6) 780 (48) 632 (39) 104 (6) 209 (13) 615 (38) 426 (26) 261 (16) 299 (18)	100 (6) 146 (9) 414 (26) 315 (20) 108 (7) 59 (4) 426 (27) 109 (7) 35 (2) 65 (4)	18 (3) 163 (25) 117 (18) 42 (7) 38 (6) 18 (3) 107 (17) 63 (10) 32 (5) 34 (5)	276 (16) 1389 (80) 427 (25) 195 (11) 90 (5) 66 (4) 469 (27) 81 (5) 72 (4) 69 (4)	<0.001 <0.001 <0.001 <0.001 0.233 <0.001 <0.001 <0.001 <0.001 <0.001
Hospital events - Anemia - Acute pancreatitis - GI bleeding - Cardiac arrest - Hypernatremia - Hyponatremia - Hyperkalemia - Hypokalemia - Acidosis - Alkalosis - Hypoglycemia - Sepsis	1898 (34) 292 (5) 689 (12) 187 (3) 171 (3) 1992 (36) 1072 (19) 442 (8) 1370 (25) 89 (2) 110 (2) 640 (12)	487 (30) 49 (3) 123 (8) 46 (3) 44 (3) 342 (21) 333 (21) 81 (5) 286 (18) 19 (1) 39 (2) 92 (6)	573 (36) 65 (4) 168 (11) 36 (2) 47 (3) 607 (38) 362 (23) 97 (6) 357 (23) 32 (2) 30 (2) 114 (7)	169 (26) 59 (9) 172 (27) 67 (10) 38 (6) 241 (38) 80 (12) 30 (5) 274 (43) 14 (2) 15 (2) 316 (49)	669 (39) 119 (7) 226 (13) 38 (2) 42 (2) 802 (46) 297 (17) 234 (14) 453 (26) 24 (1) 26 (2) 118 (7)	<0.001 <0.001 <0.001 <0.001 <0.001 <0.001 <0.001 <0.001 <0.001 0.14 0.26 <0.001
Organ Dysfunction - Circulatory failure - Respiratory failure - Neurological failure - Hematological failure	1069 (19) 918 (17) 447 (8) 1985 (36)	223 (14) 142 (9) 123 (8) 382 (24)	244 (15) 88 (6) 115 (7) 538 (34)	351 (55) 588 (92) 88 (14) 330 (51)	251 (15) 100 (6) 121 (7) 735 (43)	<0.001 <0.001 <0.001 <0.001
Treatments - Renal replacement therapy - Non-invasive ventilation - Mechanical ventilation - Blood transfusion - Nutritional support - TIPS	1119 (20) 94 (2) 573 (10) 1761 (32) 175 (3) 54 (1)	232 (14) 24 (1) 35 (2) 319 (20) 23 (1) *	283 (18) 22 (1) * 525 (33) 37 (2) 19 (1)	297 (46) 32 (5) 504 (79) 372 (58) 78 (12) 15 (2)	307 (18) 16 (1) 26 (2) 545 (32) 37 (2) 12 (1)	<0.001 <0.001 <0.001 <0.001 <0.001 <0.001
DNR status	379 (7)	114 (7)	88 (6)	35 (5)	142 (8)	0.01
Palliative consult	748 (13)	244 (15)	173 (11)	76 (12)	255 (15)	0.001
In-hospital mortality	1856 (33)	545 (34)	392 (25)	449 (70)	470 (27)	<0.001

Abbreviations: DNR: Do not resuscitate, GI: Gastrointestinal, NAFLD: Nonalcoholic fatty liver disease, TIPS: Transjugular intrahepatic portosystemic shunts. * The number is below the cut-off allowed to be reported per HCUP/NIS regulations.

## Data Availability

Data are available upon reasonable request to the corresponding author.

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
