# Peer review of "Distinct Subtypes of Hepatorenal Syndrome and Associated Outcomes as Identified by Machine Learning Consensus Clustering"

_diseases, 2023, doi:10.3390/diseases11010018_

Round 1
Reviewer 1 Report
Tangpanithandee et al found clinical subtypes of Hepatorenal syndrome using an unsupervised learning algorithm. They used a consensus clustering algorithm to find robust clusters in datasets, and the results are interesting from a clinical perspective.
I have one suggestion for making this paper clinically relevant.
The statistical comparison of the clusters identified some interesting differences between the groups. This in itself is not interesting as the clusters by definition will differ on the features used for clustering. To make the results compelling they should replicate the results in an independent dataset. The dataset used to define the clusters are patients from 2003 to 2013. They could use the data from another timeframe as test set. If there aren't enough samples for clustering, a classification algorithm could be devised using the current clusters as learning set. It will be interesting if the clinical characteristics associated with the clusters persist in the test dataset.
Author Response
Response to Reviewer#1
Tangpanithandee et al found clinical subtypes of Hepatorenal syndrome using an unsupervised learning algorithm. They used a consensus clustering algorithm to find robust clusters in datasets, and the results are interesting from a clinical perspective.
I have one suggestion for making this paper clinically relevant.
The statistical comparison of the clusters identified some interesting differences between the groups. This in itself is not interesting as the clusters by definition will differ on the features used for clustering. To make the results compelling they should replicate the results in an independent dataset. The dataset used to define the clusters are patients from 2003 to 2013. They could use the data from another timeframe as test set. If there aren't enough samples for clustering, a classification algorithm could be devised using the current clusters as learning set. It will be interesting if the clinical characteristics associated with the clusters persist in the test dataset.
Response: The reviewer raises very important point. We agree with your suggestion that cluster analysis should additionally be validated in independent testing dataset to validate the machine learning model. We currently in the process of building cohort of HRS at Mayo Clinic and additionally validate our machine learning clustering model. We have included these important information in the discussion as suggested. We also additionally included computational framework employed in the manuscript, software and packages utilized, features of the computer used in this revised manuscript.
“In addition, future studies are required to additionally validate this machine learning in different datasets of HRS patients. Lastly, while unsupervised ML clustering approach cam identify distinct phenotypes of HRS patients with differing posttransplant outcomes, ML clustering algorithms have limitations that do not directly generate risk prediction for each individual. Future studies assessing the utilization of supervised ML prediction models for outcomes among patients with HRS are needed. Nevertheless, the findings from ML clustering approach provide additional understanding towards individualized medicine for patients with HRS.”
Reviewer 2 Report
Report on the manuscript "diseases-2109680" entitled "Distinct Subtypes of Hepatorenal Syndrome and Associated Outcomes as Identified by Machine Learning Consensus Clustering"
This manuscript provides a methodology to identify clusters of hospitalized patients for hepatorenal syndrome using an unsupervised machine learning approach. Experimental results are provided. Conclusions about the present investigation are reported.
In general, I have a reasonable opinion about this work, which is relatively well written, its topic should be of interest for Diseases and its results are a suitable complement to the existing works. However, I think several aspects must be improved before it is suitable for publication, which I detail in the next section as specific comments.
1. The manuscript needs to be proofread by the authors. I have noted some drafting problems.
2. Words in the title are not usually in the keywords. In addition, the keywords are often written in alphabetical order.
3. The authors must check the use of all acronyms, abbreviations, and notations employed in the whole manuscript. At the end of Section 1, a description of the sections of the article must be added.
4. To the best of my knowledge, a good body of work has been done in the literature on this topic. The novelty and contribution of this study must be clearly stated. The authors must say what is novel in their investigation from the theoretical, methodological, and applied points of view. Are all of them or only in the application? The bibliographical review must be improved.
5. The data source must be detailed and more content about the data set must be added.
6. The authors must provide more details about the computational framework employed in the manuscript. For example, software and packages utilized, features of the computer used, runtimes, and other computational aspects must be added.
7. The authors must summarize their methodology in an algorithm in a flowchart so readers can follow it more accessible. Thus, the practitioners could have some guidelines when applying this methodology.
8. I do not have each numerical result in detail. I recommend the authors check them.
9. In my opinion, the implications and results of the study are underdeveloped and must be improved and explained further in the final section.
10. The authors must check whether all references are cited and whether all citations are in the reference list. The authors should try to cite more papers published in "Diseases" to attract the attention of our target audience.
Author Response
Response to Reviewer#2
Report on the manuscript "diseases-2109680" entitled "Distinct Subtypes of Hepatorenal Syndrome and Associated Outcomes as Identified by Machine Learning Consensus Clustering"
This manuscript provides a methodology to identify clusters of hospitalized patients for hepatorenal syndrome using an unsupervised machine learning approach. Experimental results are provided. Conclusions about the present investigation are reported.
In general, I have a reasonable opinion about this work, which is relatively well written, its topic should be of interest for Diseases and its results are a suitable complement to the existing works. However, I think several aspects must be improved before it is suitable for publication, which I detail in the next section as specific comments.
Response: Thank you for reviewing our manuscripts and your critical evaluation.
Comment #1
The manuscript needs to be proofread by the authors. I have noted some drafting problems.
Response: The manuscript was edited and proofread by one of the co-authors, who is English native speaker. We also comprehensively re-reviewed the revised manuscript and made correction as suggested.
Comment #2
Words in the title are not usually in the keywords. In addition, the keywords are often written in alphabetical order.
Response: We appreciate the reviewer’s input. The keywords were ordered in alphabetical order as suggested.
Comment #3
The authors must check the use of all acronyms, abbreviations, and notations employed in the whole manuscript. At the end of Section 1, a description of the sections of the article must be added.
Response: We appreciate the reviewer’s important comment. All acronyms, abbreviations, and notatin were checked throughout the manuscript.
Comment #4
To the best of my knowledge, a good body of work has been done in the literature on this topic. The novelty and contribution of this study must be clearly stated. The authors must say what is novel in their investigation from the theoretical, methodological, and applied points of view. Are all of them or only in the application? The bibliographical review must be improved.
Response: We appreciate the reviewer’s important comments. We have emphasized our novelty as suggested.
“This is the first unsupervised ML approach that specially targeted patients with HRS. Consensus clustering analysis without human intervention synthesized the pattern of four clinically distinct HRS subgroups with different outcomes, highlighting opportunities to further early identify high-risk HRS patients”
Comment #5
The data source must be detailed and more content about the data set must be added.
Response: The following statements have been added to describe the data source in more detail.
“The National Inpatient Sample (NIS) database is the largest all-payer database of hospital admissions in the United States. It comprises the unweighted data of more than 7 million hospital admissions annually from a 20% stratified sample of over 4,000 hospitals in the United States and represents 95% of hospital admission nationwide. The information contains patient demographics, principal diagnoses, and procedural codes. The database does not include individual patients, but it contains a single inpatient admission. Institu-tional review board approval was obtained (IRB number- 22-011097). Data in the NIS is publicly available and de-identified.
We used NIS data from 2003 to 2014 to construct a cohort of hospital admissions primarily for hepatorenal syndrome. We identified the diagnosis of hepatorenal syndrome based on the International Classification of Diseases, Ninth Edition (ICD-9) diagnosis code of 572.4.”
Comment #6
The authors must provide more details about the computational framework employed in the manuscript. For example, software and packages utilized, features of the computer used, runtimes, and other computational aspects must be added.
Response: We agree with the reviewer and thus we have additionally included these information as suggested.
“All analyses were performed using R, version 4.0.3 (RStudio, Inc., Boston, MA), with the packages of ConsensusClusterPlus (version 1.46.0) (Bioconductor Open-Source Soft-ware for Bioinformatics) [49] for consensus clustering analysis. CPU is a 4-core Intel Core i7-950 running at 3.07 GHz.”
Comment #7
The authors must summarize their methodology in an algorithm in a flowchart so readers can follow it more accessible. Thus, the practitioners could have some guidelines when applying this methodology.
Response: We agree with the reviewer. We have additionally created the flowchart of concensus clustering as the reviewer’s suggestion.
Figure 1. Workflow of Consensus Clustering.
Comment #8
In my opinion, the implications and results of the study are underdeveloped and must be improved and explained further in the final section.
Response: We agree with your suggestion that cluster analysis should additionally be validated in independent testing dataset to validate the machine learning model. We currently in the process of building cohort of HRS at Mayo Clinic and additionally validate our machine learning clustering model. Nevertheless, the findings from ML clustering approach provide additional understanding towards individualized medicine for patients with HRS. . We have also emphasized our novelty as suggested.
“In addition, future studies are required to additionally validate this machine learning in different datasets of HRS patients. Lastly, while unsupervised ML clustering approach cam identify distinct phenotypes of HRS patients with differing posttransplant outcomes, ML clustering algorithms have limitations that do not directly generate risk prediction for each individual. Future studies assessing the utilization of supervised ML prediction models for outcomes among patients with HRS are needed. Nevertheless, the findings from ML clustering approach provide additional understanding towards individualized medicine for patients with HRS.”
“This is the first unsupervised ML approach that specially targeted patients with HRS. Consensus clustering analysis without human intervention synthesized the pattern of four clinically distinct HRS subgroups with different outcomes, highlighting opportunities to further early identify high-risk HRS patients”
Comment #9
The authors must check whether all references are cited and whether all citations are in the reference list. The authors should try to cite more papers published in "Diseases" to attract the attention of our target audience.
Response: We agree with the reviewer. All references are checked and cited as appropriate, including more articles published in Diseases, as suggested.
Thank you for your time and consideration. We greatly appreciated the reviewer's and editor's time and comments to improve our manuscript. The manuscript has been improved considerably by the suggested revisions.
Reviewer 3 Report
The authors conducted and interesting large-scale study aiming at identifying distinct subtypes of hepatorenal syndrome and their outcomes using machine learning clustering. The results are well presented and the manuscript well written. I have only a few comments which may increase the value of the manuscript:
1. It would be important to add a paragraph about what new this study brings regarding hepatorenal syndrome. Did you identify any risk factor or list of risk factors of mortality that they have not been previously described in literature? Please clarify in the discussion section.
2. Moreover, does this clustering analysis help toward better management of hepatorenal syndrome? In other words, does this analysis provide any new evidence regarding the optimal management of patients based on the the pattern of clinical characteristics? Please add a paragraph in the Discussion section.
Author Response
Response to Reviewer#3
The authors conducted and interesting large-scale study aiming at identifying distinct subtypes of hepatorenal syndrome and their outcomes using machine learning clustering. The results are well presented and the manuscript well written. I have only a few comments which may increase the value of the manuscript:
Response: Thank you for reviewing our manuscripts and your critical evaluation.
Comment #1
It would be important to add a paragraph about what new this study brings regarding hepatorenal syndrome. Did you identify any risk factor or list of risk factors of mortality that they have not been previously described in literature? Please clarify in the discussion section.
Response: We appreciate the reviewer’s important comments and we agree with the reviewer. Thus, we additionally emphasize the important risk factors that we found and also the benefit of these analysis and clustering. The following text has been added in discussion section as suggested.
“After analyzing these four distinct clusters, we found that there are lots of risk factors that play important roles in in-hospital mortality of HRS patients, especially multiple or-gan failure, respiratory failure that required mechanical ventilation, circulatory failure leading to cardiac arrest, hematological failure that led to bleeding and needed for blood transfusion, metabolic acidosis, and renal impairment that required RRT which can be found in Cluster 3. Other risk factors are age and cardiovascular comorbidities, including hypertension and diabetes mellitus, which can be found in Cluster 1. Due to these ana-lyzed clustering and risk factors, medical providers can improve in-hospital management and also researchers can do further studies to predict outcomes of HRS patients in hospi-tal admission.”
Comment #2
Moreover, does this clustering analysis help toward better management of hepatorenal syndrome? In other words, does this analysis provide any new evidence regarding the optimal management of patients based on the the pattern of clinical characteristics? Please add a paragraph in the Discussion section.
Response: Thank you for reviewer’s comment. We added the text as in our previous response in comment #1 as suggested.
Thank you for your time and consideration. We greatly appreciated the reviewer's and editor's time and comments to improve our manuscript. The manuscript has been improved considerably by the suggested revisions.
Round 2
Reviewer 2 Report
Accept in present form.